# Evidence on access to healthcare information by women of reproductive age in low- and middle-income countries: Scoping review

Joyce Twahafifwa Shatilwe[1]*, Desmond Kuupiel[1], Tivani P. Mashamba-Thompson[1,2]

1 Discipline of Public Health Medicine, School of Nursing and Public Health, University of KwaZulu-Natal, Durban, South Africa, 2 Faculty of Health Sciences, University of Pretoria, Pretoria, South Africa

☙ These authors contributed equally to this work.
* jshatilwe@hotmail.com

**Data Availability Statement:** All relevant data are within the paper and its Supporting Information files.

## Abstract

### Background

A majority of women of reproductive age in low- and middle-income countries (LMICs) are not able to access healthcare information due to different factors. This scoping review aimed to map the literature on access to healthcare information by women of reproductive age in LMICs.

### Methods

The literature search was conducted through the following databases: Google Scholar, Science Direct, PubMed, EBSCOhost (Academic search complete, CINAHL with full text, MEDLINE with full text, MEDLINE, and PsycINFO), Emerald, Embase, published and peer-reviewed journals, organizational projects, reference lists, and grey literature.

### Results

A total of 377 457 articles were identified from all the databases searched. Of these, four articles met inclusion criteria after full article screening and were included for data extraction. The themes that emerged from our study are as follows: accessibility, financial accessibility/affordability, connectivity, and challenges. This study demonstrated that there are minimal interventions that enable women of reproductive age to access healthcare information in terms of accessibility, financial accessibility, and connectivity.

### Conclusion

The findings of the study revealed poor access and utilization of healthcare information by women of reproductive age. We, therefore, recommend primary studies in other LMICs to determine the accessibility, financial accessibility, connectivity, and challenges faced by women of reproductive age in LMICs.

**Funding:** The author(s) receive no specific funding for this work.

**Competing interests:** The authors have declared that no competing interests exist.

**Abbreviations:** AFHS, Adolescent-friendly health services; BREC, Bio-Medical Research Committee; LMICs, Lower- and middle-income countries; MHC, Maternal and child health; MMAT, Mixed Method Appraisal Tool; NCDs, Non-communicable diseases; PCC, Participants-Concept-Context; PICOS, Population, Intervention, Comparison, Outcomes, and Study Setting; PRISMA-ScR, Preferred Reporting Items for Systematic Review Extension for Scoping Review; SDG, Sustainable Development Goal; STATA, Statistical Analysis Software; UHC, Universal Health Coverage; UNFPA, United Nations Population Fund; UKZN, University of KwaZulu-Natal; WHO, World Health Organization.

# Background

There are more than a billion people around the globe, mainly in LMICs, who are unable to access essential healthcare information due to a variety of reasons [1]. Roughly, about 150 000 women in Africa die each year from causes related to pregnancy and childbearing and the risk of dying from maternal-related causes for African women is in the range of one in 25 [2]. Barriers to accessing and utilizing healthcare information have been classified into access, availability, acceptability, cultural and traditional preferences, confidence in care and quality of services, health awareness and knowledge, and affordability [3].

Access is part of universal health coverage (UHC) components (access, effective coverage, and need) [4]. According to the World Health Assembly, UHC is defined as "All people receiving comprehensive quality health services they need without enduring financial costs in so doing thereby achieving equity in access" [4]. Access has four dimensions, namely: geographic accessibility, availability, financial accessibility, and acceptability [5]. Women of reproductive age will utilize maternal and child healthcare information (MCHI) to their maximum if they have access to healthcare services [1]. Possible factors influencing the utilization of healthcare information, either public or private, are socio-economic factors, and cultural beliefs and practices. In particular, the healthcare system itself, the distance from health facilities, availability, affordability, and quality of healthcare information are among the prevalent factors that influence healthcare utilization [1].

Improved accessibility to healthcare information has been and continues to be, a central objective of health policy for optimum health system performance [6, 7]. Substantial research has been conducted focusing mainly on access to healthcare information and sustained attention was given mostly to accessibility issues in health policies and the context of health services research; however, community and policymakers continue to seek answers to this fundamental question [7]. A key element of UHC is that of ensuring access to and use of needed healthcare information for everyone. This can only be achieved if accessibility to healthcare information is identified and utilized by women of reproductive age [8]. The main aim of this study was to evaluate the accessibility to healthcare information for women of reproductive age in low and middle–income countries (LMCI). This was done by undertaking a literature search for available interventions which enable reproductive age women in LMIC access healthcare information. Evidence on the topic area will be needed to guide the study on the available scientific knowledge that was collected and how far it was utilized.

# Materials and methods

This manuscript is part of the main study approved by the UKZN Bio-Medical Research Ethical Committee. The main study was approved with a written consent. A systematic scoping review protocol was published in *BMC* journal under the title: Mapping evidence on access to healthcare information by women of reproductive age in low- and middle-income countries: scoping review protocol [9]. A scoping review was selected in this study as the most appropriate method to map literature on evidence on access to healthcare information by women of reproductive age in LMICs. The scoping review tried to search different interventions/strategies in place that enable women of reproductive age in low-and middle-income countries to access healthcare information. The interventions are such as health promotion interventions programmes, health outreach programmes, facility based education initiative, health education initiatives (comprehensive sexuality education programmes), programmes to scale up healthcare information technology to promote technology (text messages, mobile health (M-health), community based outreach programmes and school health programmes. The scoping review

was guided by Tricco et al (2018) and followed the Preferred Reporting Items for Systematic Reviews and Meta-Analyses Extensions for Scoping Reviews (PRISMA-ScR) protocol. The following are the components of PRISMA review protocol: title; abstract; introduction (rationale and objectives); methods (protocol and registration, eligibility criteria, information sources, search, selection of sources of evidence, data charting process, data items, critical appraisal of individual sources of evidence and synthesis of results); results (selection of sources of evidence, characteristics of sources of evidence, critical appraisal within sources of evidence, results of individual sources of evidence, synthesis of results); discussion (summary of evidence); limitations; conclusion and funding. The results of the review were presented according to PRISMA-ScR [10]. Table 1 below indicates the PRISMA-ScR checklist.

## Eligibility criteria

The study was guided by inclusion and exclusion criteria to develop the research questions, to ensure the correct identification and selection of relevant studies.

**Inclusion criteria.**   Included studies met the following criteria:

- Women of reproductive age (14 to 49 years old).

- Evidence from the period 2004 until the present.

- The study conducted in lower- and middle-income countries.

- Studies that focus on interventions enabling access to healthcare information.

- Articles including peer-reviewed journal articles, grey literature, and primary studies.

- The scoping review will include studies of all study designs.

**Exclusion criteria.**   Publications that were excluded included:

- Studies that involved women below 14 years old and above 49 years old.

- Studies that did not focus on healthcare information.

- Studies that were done earlier than in 2004.

- Studies that reported on non-health information.

- Systematic, scoping, expert and literature reviews.

## Information sources

The study was conducted by a team with expertise in content and methodology that ensured the successful completion of the study. The data were retrieved from the following databases: Google Scholar, Science Direct, PubMed, EBSCOhost (Academic search complete, CINAHL with full text, MEDLINE with full text, MEDLINE, and PsycINFO), Emerald, Embase, and Cochrane Database of Systematic Review (CDSR). Reference lists for included studies, conferences, and websites were searched for relevant studies. The following keywords were applied to search for eligible studies: Interventions, Access, Healthcare information, women of reproductive age, Low- and Middle-Income Countries.

## Search strategy

A comprehensive search strategy was conducted in consultation with the University of KwaZulu-Natal School of Nursing and Public Health Medicine librarian. We searched and

**Table 1. Preferred reporting items for systematic reviews and meta-analyses extensions for scoping reviews (PRISMA-ScR) checklist.**

| SECTION | ITEM | PRISMA-ScR CHECKLIST ITEM | REPORTED ON PAGE # |
|---|---|---|---|
| **TITLE** | | | |
| TITLE | 1 | **Evidence on access to healthcare information by women of reproductive age in low- and middle-income countries: Scoping review** | 66 |
| **ABSTRACT** | | | |
| Structured summary | 2 | Background: A majority of women of reproductive age in low- and middle-income countries (LMICs) are not able to access health services due to different factors. The main objective of this scoping review is to map the literature on access to healthcare information by women of reproductive age in LMICs. | 67 |
| | | Methods: The literature search was conducted through the following databases: Google Scholar, Science Direct, PubMed, EBSCOhost (Academic search complete, CINAHL with full text, MEDLINE with full text, MEDLINE, and PsycINFO), Emerald, Embase, published and peer-reviewed journals, organizational projects, reference lists, and grey literature. | |
| | | Results: A total of 377 457 articles were identified from all the databases searched. Of these, four articles met inclusion criteria after full article screening and were included for data extraction. The themes that emerged from our study are as follows: accessibility, financial accessibility/affordability, connectivity, and challenges. This study demonstrated that there are minimal interventions that enable women of reproductive age to access healthcare information in terms of accessibility, financial accessibility, and connectivity. | |
| | | Conclusion: The findings of the study revealed poor access and utilization of MCHI by women of reproductive age. We, therefore, recommend primary studies in other LMICs to determine the accessibility, financial accessibility, connectivity, and challenges faced by women of reproductive age in LMICs. | |
| | | Keywords: Access, healthcare information, women of reproductive age, low-and middle-income countries | |
| **INTRODUCTION** | | | |
| Rationale | 3 | This scoping review maps the available literature on access to healthcare information by women of reproductive age in LMICs. It also provides a general overview on what factors contribute to women of reproductive age in LMICs not to access healthcare information. | 68 |
| Objectives | 4 | To map the evidence on interventions aimed at enabling access to health information in LMICs. | 69 |
| **METHODS** | | | |
| Protocol and registration | 5 | Not registered | |
| Eligibility criteria | 6 | **Inclusion criteria** | 71–72 |
| | | Included studies met the following criteria: | |
| | | • Women of reproductive age (14 to 49 years old). | |
| | | • Evidence from the period 2004 until the present. | |
| | | • The study conducted in lower- and middle-income countries. | |
| | | • Studies that focus on interventions enabling access to healthcare information. | |
| | | • Articles including peer-reviewed journal articles, grey literature, and primary studies. | |
| | | • All studies to be included irrespective of their study designs. | |
| | | **Exclusion criteria** | |
| | | Publications that were excluded included: | |
| | | • Studies involving women below 14 years old and above 49 years old. | |
| | | • Studies that did not focus on healthcare information. | |
| | | • Studies were done earlier than in 2004. | |
| | | • Studies reporting on non-health information. | |
| | | • Systematic, scoping, expert and literature reviews. | |
| Information sources | 7 | A team with content and methodological expertise was assembled to ensure the successful completion of the study. The data were retrieved from the following databases: Google Scholar, Science Direct, PubMed, EBSCOhost (Academic search complete, CINAHL with full text, MEDLINE with full text, MEDLINE, and PsycINFO), Emerald, Embase, and Cochrane Database of Systematic Review (CDSR). Reference lists for included studies, conferences, and websites were searched for relevant studies. The following keywords were applied to search for eligible studies: Interventions, Access, Healthcare information, Healthcare services, Low- and Middle-Income Countries. | 71 |
| Search | 8 | A comprehensive search strategy was conducted in consultation with the University of KwaZulu-Natal School of Nursing and Public Health Medicine librarian. We searched and assessed papers containing "access" and "healthcare information", either in the title, abstract, or body. We searched further, using keywords such as "women of reproductive age" or "lower- and middle-income countries". The study was guided by inclusion and exclusion criteria to develop the research questions, to ensure the correct identification and selection of relevant studies. Studies were selected according to the PCC framework recommended by the Joanna Briggs Institute for scoping reviews, as stipulated in Table 2 [10, 11]. | 69 |
| | | The LMICs in the context was determined by the World Bank list of economies [12] which classifies countries according to their economic status, and the age range of reproductive women used is classified by the WHO as 14 to 49 years old [13]. The study's focus is on the period from 2004 to the present. Focusing on a 15-year period enabled us to collect broader knowledge from studies generated during the specified period. The study included articles written in all languages and used in the University of KwaZulu-Natal Systematic Review Services to help with searching for interpreters in cases of retrieved studies published in other languages. | |

*(Continued)*

**Table 1.** (Continued)

| SECTION | ITEM | PRISMA-ScR CHECKLIST ITEM | REPORTED ON PAGE # |
|---|---|---|---|
| Selection of sources of evidence | 9 | Study selection occurred in three stages. The first stage was conducted by one reviewer through title screening databases guided by the eligibility criteria. After the title screening was completed, the process continued with abstract and full article screening. These two processes were conducted by two independent reviewers by following the inclusion and exclusion criteria. A screening form was developed guided by the questions derived from the eligibility criteria. The form was used to guide the abstract and full article screening process. During the abstract screening stage, differences in reviewer responses were resolved through a discussion by the review team until consensus was reached. Discrepancies at the full-text screening stage were resolved by involving a third screener. | 73 |
| Data charting process | 10 | The review team collectively developed the data charting form and determined suitable variables to be extracted to help answer the study research questions. The data extraction form was piloted by two independent reviewers (JS & TPMP). It was then adjusted accordingly. The review team then extracted data from included studies using the following domains: Author and date; study title; study aim; study design; study setting (country); geographic setting (rural/urban); study population, age, and female percentage; interventions implemented/utilized; intervention type; intervention duration; key findings; significant findings; and study conclusion. | 72 |
| Data items | 11 | Accessibility, Financial accessibility/affordability, Connectivity and Challenges | 79–81 |
| Critical appraisal of individual sources of evidence | 12 | The risk of bias was assessed in the included studies guided by the MMAT 2018 version. For each included study, an appropriate category of studies was used for appraisal by looking at the description of the methods used. All the categories included (qualitative, quantitative, and mixed-method) had five criteria. The responses were either "yes" or "no" or "can't tell". A detailed presentation of the ratings of each criterion to inform the quality of the included studies was presented. The studies were assessed based on their method: qualitative, quantitative, and mixed methods. For qualitative studies, the following areas were assessed: appropriateness of the research question and problem, adequacy of the data collection method and the form of data, data analysis used, sufficient interpretation of the results, coherence between qualitative data sources, collection, analysis, and interpretation. For quantitative studies, the following areas were assessed: Relevance of sampling strategy, the match between respondents and the target population, appropriateness of measurements, risk of no response, and appropriateness of statistical analysis to answer research questions. For mixed-method studies, the following areas were assessed: Reason for conducting a mixed-method, information on the integration of qualitative and quantitative phases and results, results brought together into overall interpretation, adequately addressed divergences, adequately addressed inconsistencies between quantitative and qualitative results, and adherence to the quality criteria of each tradition of methods involved. | 73 |
| Synthesis of the results | 13 | First, we presented the characteristics of the included studies in the following domains: The number of studies reporting on the specific outcome, the country where the study was conducted, participants in the study, the aim of the study, the outcomes and the research or practice gaps revealed for that particular outcome. A table was produced with the following domains: Author and date, study title, study design, study settings, geographic setting, study population, age, and female percentage. | 73 |
| | | Second, the literature was organized thematically, based on the grounded themes extracted from the studies. The themes were as follows: Accessibility, financial accessibility, connectivity, and challenges. Each theme was discussed separately. The results were presented using PRISMA-ScR. | |
| **RESULTS** | | | |
| Selection of sources of evidence | 14 | As shown in Fig 1, a total of 377 457 articles were identified after the database search; 376 707 articles were found ineligible and removed. Only 750 articles were found eligible. A further 76 articles were found to be ineligible and 65 duplicates were removed; thus, 609 articles were found eligible for title screening. Subsequently, 408 studies were excluded and 201 abstracts were then screened. Of these, 51 were selected for full-text article screening. Following full-text article screening, a total of 48 articles were excluded and only four met inclusion criteria and were included for data extraction (See Fig 1). | 73 |
| Characteristics of sources of evidence | 15 | Table 2 depicts the characteristics of the included studies. All the eligible studies were published from the year 2004 to the present. The study setting for all the included studies was LMICs. All of the included primary studies showed evidence on access to healthcare information by women of reproductive age in LMICs. Two of the four included studies were conducted in rural settings [59, 60] and the other two were conducted in both rural and urban settings [61, 62]. The included studies were conducted in the following countries: One in Myanmar [61], one in South Africa [62], one in Eastern Uganda [60], and one in Nepal [59]. The total sample size from included studies was 11 134 participants and all were women. All studies focused on female participants [59–62]. Of the four included studies two were quantitative studies [60, 61], one was mixed methods [59] and one was a qualitative study [62]. One of the four included studies reported on universal health coverage [61], another reported on value for money of mobile maternal health information messages [62], two reported on increasing access to maternal healthcare services and exploring the role of telemedicine, respectively [59, 60]. | 77 |
| Critical appraisal with sources of evidence | 16 | All included studies underwent methodological quality assessment using MMAT version 2018 [16]. The overall percentage quality score was calculated for included studies. The scores ranged from 71.4% to 100%. They were interpreted as follows: Below 51% low quality, 51–75% average quality and 76–100% high quality. | 79 |
| Result of individual sources of evidence | 17 | **Accessibility** | 79–81 |
| | | Out of the four included studies, one study reported on the accessibility of maternal healthcare to pregnant women. The study conducted in Myanmar reported on universal health coverage. Its main purpose was to determine the national and subnational health service coverage and financial risk protection [61]. Twenty-six health service indicators were examined by using nationally representative data from the Myanmar Demographic and Health Survey (2016) and the Integrated Household Living Condition Assessment (2010). The same study also assessed the incidence of catastrophic health payment and impoverishment caused by out-of-pocket payments [61]. The study findings show that nationally, the coverage of health service indicators ranged from 18.4% to 96.2%. The findings further indicate that the coverage of most health services indicators did not reach the universal health coverage of 80% [61]. Also found is that increased levels of education (either the mothers or partners) have a positive influence on the access to perinatal care services [61]. The study shows that women with some higher education were likely to attend at least four ANC visits and to deliver in the health facilities compared to those with no education [61]. The study recommends that to achieve 80% coverage, efforts should focus on the expansion of services and increased coverage to reduce the gap that exists in maternal, neonatal, and child health coverage [61]. Although the study mentioned the gap in maternal, neonatal, and health coverage, a research gap still exists to explore interventions that can be used to attract women of reproductive age to use maternal healthcare services. | |

(*Continued*)

**Table 1.**  (Continued)

| SECTION | ITEM | PRISMA-ScR CHECKLIST ITEM | REPORTED ON PAGE # |
|---|---|---|---|
| | | **Financial accessibility/affordability** | |
| | | One study reported on the affordability of maternal healthcare services to women of reproductive age. Mayora et al. (2014) conducted a study in eastern Uganda to assess the influence of demand- and supply-side programmes on increasing access to maternal health services [60]. This was a costing study that used vouchers. Costs were based on market prices as recorded in programme records [60]. Pregnant mothers were issued vouchers that they needed to present to the service provider, who, in return, would be reimbursed by the research team on presenting the voucher (Dec 2009 to March 2010 and June 2010 to June 2011) [60]. The outcome of this study revealed that transport vouchers scooped the highest, followed by health system strengthening, while maternal services vouchers were the lowest [60]. The study further showed that the average cost of transport per women to and from the health facility was US$4.60. It also indicated that delivery cost was the highest at US$ 317 157, followed by ANC at US$ 107 890, while postnatal care cost the least at US$ 7.60 [60]. The findings also revealed that when subsidizing maternal healthcare costs through demand and supply, side initiatives are a lesser cost and would require fewer resources than expected [60]. Although the study indicated that a voucher study can be used and may not require a significant amount of resources, an extensive research study still needs to be conducted on its sustainability and what would be a reasonable cost to the entire population. | |
| | | **Connectivity** | |
| | | Two studies reported evidence on the connectivity of women to maternal healthcare services [59, 62]. A study conducted in Nepal reported on telemedicine for improving access to healthcare services by women and girls in rural Nepal [59]. The purpose of the study was to assess the influence of telemedicine in reducing gender-based challenges that women and girls are facing to reach healthcare services in rural areas of Nepal [59]. The sample for this study were women and girls who used video conference-based telemedicine services before January 2015 and those who received mobile phone-based telemedicine in January of the same year [59]. The results of the study revealed that telemedicine positively influences travel restrictions, treatment expenses, and apprehension regarding sexual and reproductive health consultations. It was further shown that this intervention decreased travel time, because of timely assistance to access healthcare services for women and girls. At the same time, it created the convenience in improved time management so that the women would still be able to perform household chores and other activities [59]. This study revealed that telemedicine, especially mobile phone-based telemedicine, encouraged women and girls to have the confidence and freedom to ask about sexual health-related information from a doctor located at a far distance [59]. Besides, because of the confidentiality insurance of mobile phone-based telemedicine, fear, or timidity is reduced because the identity of the women and girls is not revealed [59]. A study conducted in Gauteng in South Africa that aimed at modeling the incremental cost-effectiveness of gradually scaling up text messaging services to pregnant women showed this as a cost-effective strategy for bolstering ANC and childhood immunizations [62]. The two studies indicated modalities of accessing services by women of reproductive age from a distance. However, there is still a research gap in how to access the population of women of reproductive age who are not within the diameter of network coverage and those who cannot afford to acquire technology devices such as cell phones. | |
| | | **Challenges** | |
| | | Two studies presented evidence on challenges or barriers to accessing maternal healthcare services by women of reproductive age [59, 61]. Study findings by Han et al. (2018) highlighted certain hindrances to effective implementation of maternal healthcare programmes, which include heavy workloads, geographical and transportation barriers, poor supervision and training, and insufficient replacement of auxiliary midwife kits [61]. The lack of accessible healthcare facilities, inadequate health workforce, and health budget allocation were the main causes of the regional inequity [61]. The main barriers in most Asia-Pacific countries, including Myanmar, are high use fees and cash payment for healthcare services, which are highly likely to hinder disadvantaged communities from accessing healthcare facilities [61]. Maternal, neonatal, and child health (MNCH) indicators such as postnatal care for neonates and institutional delivery recorded the lowest coverage [61]. Besides, financial constraints and lack of transportation are the prevalent hindrances to accessing delivery care facilities [61]. | |
| | | A study by Parajuli and Doney (2017) reported that even though telemedicine is inclined to reduce gender-based barriers for women and girls, we should take note that their capacity to benefit from telemedicine is limited, mainly in two ways. Firstly, women and girls who have no mobile phone find it difficult to call a remote doctor. Secondly, women with lower levels of education had to be assisted to utilize mobile phone-based telemedicine [59]. Although many challenges that may hinder women of reproductive age to access maternal healthcare services have been outlined, a research gap exists on how these challenges can be alleviated to empower women of reproductive age to access maternal healthcare services. | |
| Synthesis of result | 18 | The themes that emerged from the four included studies are as follows: Accessibility, financial accessibility, connectivity and challenges. All four studies showed evidence on access to healthcare information by women of reproductive age in LMICs. | 79 |
| **Discussion** | | | |
| Summary of evidence | 19 | This scoping review mapped the available literature on access to healthcare information by women of reproductive age in LMICs. It also provided a general overview of what factors contribute to women of reproductive age in LMICs not accessing healthcare information. The study revealed that that there was lack of literature in this area. Evidence provided information on the following themes: Accessibility, financial accessibility, connectivity, and challenges being faced by women of reproductive age to accessing healthcare services. It was also shown that women with high education have greater access to healthcare information than those with lower education. Furthermore, it was revealed that MNCH gaps exist. It also provided evidence on demand- and supply-side initiatives such as transport vouchers and maternal healthcare services vouchers. The findings of the study showed incremental cost-effectiveness of exposure to SMS text messages during the provision of maternal healthcare services, which may increase access to healthcare information. Evidence also showed that telemedicine reduces travel restrictions, treatment expenses, and apprehension regarding sexual and reproductive healthcare information. | 82–84 |
| | | The findings of this study showed that access to health services indicators was below the UHC, which, in turn, affects access to healthcare information. Similar to our findings, a study done in Nigeria by Aregbeshola et al. (2017) found that about 10–20% of the monthly household income was spent on healthcare by 46.8% of their respondents. It further found that a total of 97.9% of respondents had no health insurance coverage [33]. Another study conducted in Nigeria by Okoronkwo et al. (2015) found that the costs of medical treatment and not having insurance coverage was a major financial barrier to utilization and treatment services [61]. These two studies are in agreement with our findings in LMICs. | |

(*Continued*)

**Table 1.** (Continued)

| SECTION | ITEM | PRISMA-ScR CHECKLIST ITEM | REPORTED ON PAGE # |
|---|---|---|---|
| | | Similarly, a high-income country study conducted in New Jersey in the United States by Holstein et al. (2017) revealed that patient access to care under ten large insurance plans varied by the plan, but overall, access was difficult. Furthermore, maternal healthcare is said to be more accessible for women with high levels of education, and therefore have a greater chance of accessing healthcare information compared to those with low levels of education. This could be because educated women are generally more exposed to more information than non-educated women. A study by Vidler et al. (2016) found poor education to be one of the hindering factors in accessing maternal healthcare services [62]. Contrary to our findings, a study conducted in Pakistan in 2017 found that distance, transport, staff availability, income, service hours, and service organization are some of the barriers to maternal healthcare services where healthcare information can be accessed [63]. Furthermore, the existing MNCH gap that our study revealed could be due to the challenges or barriers that hinder access to maternal healthcare services in low research settings. Illustrating this, Asghari et al. (2018), in a study in urban slums of Lagos in Nigeria, found that a total of 80.3% of their respondents had an estimated travel distance ranging from 6 to 10 km to reach a healthcare facility [33]. Other barriers were revealed by a study conducted in Nepal by Paudel et al. (2018) where the low focus of primary healthcare on engagement and empowerment was responsible evident in two areas; firstly, in quality of care: poor acceptance, feeling unsafe and uncomfortable in health facilities; and secondly, in health governance: failure in delivering healthcare services during pregnancy and delivery were some of the challenges identified [64]. | |
| | | Our study found that demand- and supply-side initiatives such as transport vouchers and maternal healthcare services vouchers were effective and may not require a significant amount of resources. However, contrary to our findings, a study conducted in Ghana on fee-free maternal healthcare services found that direct costs associated with ANC in the public healthcare facilities were still a significant barrier to pregnant women who wanted to utilize services from these facilities [46]. Contrary to our findings, a study conducted in India by Sahoo et al. (2017) found that service providers also experienced barriers that may hinder service provision, such as physical access to facilities [39]. Another study similar to ours conducted in Bangladesh by Wahed et al. (2017), on sex worker access to sexual and reproductive healthcare services, revealed that financial problems, shame about receiving care, the unwillingness of service providers to provide care, unfriendly behaviour of the providers and distance to care were some of the challenges that prevent them from receiving sexual and reproductive healthcare services [44]. Furthermore, the findings of these studies showed incremental cost-effectiveness of exposure to SMS text messaging during the provision of maternal healthcare services. This is in agreement with a 2016 systematic study finding that showed similar incremental cost-effectiveness of exposure to SMS text messaging during the provision of maternal healthcare services [6]. Our findings were also in agreement with a study conducted in Afghanistan by Yamin and Kaewkungwa (2018), showing that 81.7% of their participants were willing to receive health messages via a mobile phone. The same study also revealed that the automated voice call was the most preferred method for sending health messages. More than 90% of women are willing to receive reminders for their children's vaccination and ANC [65]. Evidence also showed that telemedicine reduced travel restrictions, treatment expenses, and apprehension regarding sexual and reproductive health. A high-income country study by Jandovitz et al. (2018) which was conducted with organ transplantation patients was in agreement with our findings that telemedicine has the potential to improve the healthcare delivery model by providing increased patient to healthcare team interactions and access, which optimizes engagement and outcomes [66]. | |
| | | The SDG target 3.7 focuses on the universal access to sexual and reproductive healthcare services, achievable by 2030 [67]. The WHO recommendations on health promotion interventions for maternal and newborn health stipulates twelve recommendations to strengthen maternal and newborn health. Two of the recommendations are interventions to promote awareness of human, sexual and reproductive rights and the right to access quality skilled care; and community mobilization through facilitated participatory learning and action cycles, highlighting women's groups to empower women with relevant information and knowledge to access healthcare services. | |
| Limitations | 20 | Several challenges associated with access to healthcare information were reported: the absence of accessible health facilities, an insufficient workforce, insufficient health budget allocation, high user fees, and direct out-of-pocket payment for healthcare services, financial constraints, and transport constraints [61]. Bearing in mind that many LMICs are adopting the UHC to help them achieve SDG number 3, it would be advisable and beneficial for the Ministry of Health and Social Services to explore interventions that will enable women of reproductive age to better access MCHI without suffering any hardship. Our study findings show that there is limited published literature specific to strategies in place that would enable these women to access healthcare information in general in LMICs. Therefore, we hope that this study's results will prompt further studies to provide a contextual insight for these strategies to increase the use of maternal and reproductive healthcare services. Considering that only two studies mentioned interventions that can attract women of reproductive age to access maternal healthcare services in LMICs, we would like to recommend future pilot studies and randomized control trials to access strategies aimed at enabling these women to access the healthcare information under discussion. We would like to further recommend that future intervention programmes should be developed and implemented to ensure quality and desirable maternal healthcare services outcomes. However, this study was limited in that we only included studies that were conducted between 2004 to the present, excluding those conducted before 2004. | 84–85 |
| Conclusions | 21 | This study demonstrated that some strategies might be useful to expose a large number of women from receiving MCHI without any challenges. It also indicated that there is a need for more research on evidence that would enable access to MCHI by women of reproductive age in LMICs. | 85 |
| **FUNDING** | | | |
| Funding | 22 | Not applicable | - |

assessed papers containing "interventions and healthcare information", "access" and "healthcare information", either in the title, abstract, or body. We searched further, using keywords such as "women of reproductive age" or "lower- and middle-income countries". The study was guided by inclusion and exclusion criteria to develop the research questions, to ensure the correct identification and selection of relevant studies. Studies were selected according to the PCC framework recommended by the Joanna Briggs Institute for scoping

reviews, as stipulated in Table 2 [11, 12]. The data base for the studies that was included is depicted under Table 3.

The LMICs in the context was determined by the World Bank list of economies [13] which classified countries according to their economic status, and the age range of reproductive women used is classified by the WHO as 14 to 49 years old [14]. The study's focus is on the period from 2004 to the present. Focusing on a 15-year period enabled us to collect broader knowledge from studies generated during the specified period. The study included articles written in all languages and used in the University of KwaZulu-Natal Systematic Review Services to help with searching for interpreters in cases of retrieved studies published in other languages. The Population-Concept-Context (PCC) framework for determining the eligibility of this study for the primary research question was adopted (Table 2).

## Selection of sources of evidence

Study selection occurred in three stages. The first stage was conducted by one reviewer through title screening databases guided by the eligibility criteria. After the title screening was completed, the process continued with abstract and full article screening. These two processes were conducted by two independent reviewers by following the inclusion and exclusion criteria. A screening form was developed guided by the questions derived from the eligibility criteria. The form was used to guide the abstract and full article screening process. During the abstract screening stage, differences in reviewer responses were resolved through a discussion by the review team until consensus was reached. Discrepancies at the full-text screening stage were resolved by involving a third screener.

## Data charting process

The review team collectively developed the data charting form and determined suitable variables to be extracted to help answer the study research questions. The data extraction form was piloted by two independent reviewers (JS & TPMP). It was then adjusted accordingly. The review team then extracted data from included studies using the following domains: Author and date; study title; study aim; study design; study setting (country); geographic setting (rural/urban); study population, age, and female percentage; interventions implemented/utilized; intervention type; intervention duration; key findings; significant findings; and study conclusion.

## Data items

The data charting form included a mixture of general information about the study and specific information related to the study population, the type of intervention, outcome measures employed and the study design. The information included the following: Author(s), year of publication, journal full reference, aims or research questions, participant characteristics,

**Table 2. PCC framework.**

|   | Criteria | Determinants |
|---|----------|--------------|
| P | Population | • Women of reproductive age in LMICs |
| C | Concept | • Any interventions that enable women of reproductive age to access healthcare information carried out during 2004 to the present |
| C | Context | • Research articles are limited to LMICs |
|   |          | • All languages will be included |
|   |          | • Studies conducted as from 2004 to the present will be included |

**Table 3. Database search.**

| Date | Database | Key search words | Number of articles found | Number of articles found eligible |
|---|---|---|---|---|
| 18/8/2018 | PubMed | Access and healthcare services | 84 889 | 205 |
| 18/8/2018 | EBSCOhost | Access and healthcare services | 31 825 | 90 |
| 18/8/2018 | EBSCOhost | Access and healthcare services | 30 297 | 213 |
| 18/8/2018 | Google Scholar | Access and healthcare services | 153 000 | 68 |
| 18/8/2018 | Emerald | Access and healthcare services | 74 501 | 32 |
| 19/08/2018 | EBSCOhost | Access and healthcare services and Lower and Middle Income Countries | 72 | 19 |
| 19/08/2018 | Emerald | Access and healthcare services and Lower and Middle Income Countries | 535 | 11 |
| 19/08/2018 | Google Scholar | Access and healthcare services Lower and Middle Income Countries | 2100 | 81 |
| 19/08/2018 | PubMed | Access and healthcare services and Lower and Middle Income Countries | 238 | 31 |
| | | Total | **377 457** | 750 |
| | | Articles deleted | - | 76 |
| | | | = | 674 |
| | | Duplicates removed | - | 65 |
| | | Total eligible | = | 609 |

recruitment context, sampling method, study design, theoretical background, data collection method, data analysis, intervention, intervention outcome, most relevant findings, conclusion and comments [15].

## Critical appraisal of individual sources of evidence

The risk of bias was assessed in the included studies guided by the mixed method appraisal tool (MMAT) 2018 version [16]. For each included study, an appropriate category of studies was used for appraisal by looking at the description of the methods used. All the categories included (qualitative, quantitative, and mixed-method) had five criteria. The responses were either "yes" or "no" or "can't tell". A detailed presentation of the ratings of each criterion to inform the quality of the included studies was presented. The studies were assessed based on their method: qualitative, quantitative, and mixed methods. For qualitative studies, the following areas were assessed: appropriateness of the research question and problem, adequacy of the data collection method and the form of data, data analysis used, sufficient interpretation of the results, coherence between qualitative data sources, collection, analysis, and interpretation. For quantitative studies, the following areas were assessed: Relevance of sampling strategy, the match between respondents and the target population, appropriateness of measurements, risk of no response, and appropriateness of statistical analysis to answer research questions. For mixed-method studies, the following areas were assessed: Reason for conducting a mixed-method, information on the integration of qualitative and quantitative phases and results, results brought together into overall interpretation, adequately addressed divergences, adequately addressed inconsistencies between quantitative and qualitative results, and adherence to the quality criteria of each tradition of methods involved.

## Synthesis of results

First, we presented the characteristics of the included studies in the following domains: The number of studies reporting on the specific outcome, the country where the study was

conducted, participants in the study, the aim of the study, the outcomes and the research or practice gaps revealed for that particular outcome. A table was produced with the following domains: Author and date, study title, study design, study settings, geographic setting, study population, age, and female percentage.

Second, the literature was organized thematically, based on the grounded themes extracted from the studies. The themes were as follows: Accessibility, financial accessibility, connectivity, and challenges. Each theme was discussed separately. The results were presented using PRISMA-ScR.

## Result

### Selection of sources of evidence

As shown in Fig 1, a total of 377 457 articles were identified after the database search; 376 707 articles were found ineligible and removed. Only 750 articles were found eligible. A further 76 articles were found to be ineligible and 65 duplicates were removed; thus, 609 articles were found eligible for title screening. Subsequently, 408 studies were excluded and 201 abstracts were then screened. Of these, 51 were selected for full-text article screening. Following full-text article screening, a total of 47 articles were excluded and only four met inclusion criteria and were included for data extraction (See Fig 1). Some of the reasons for exclusion were as follows: Fourteen records do not focus on the age between 14–49 years old, one records do not meet age requirement, six records focus on general healthcare, eight records report on sexual and reproductive healthcare service, 15 records report on maternal healthcare services and three were literature review studies.

A total of 48 articles were excluded after full article screening. Reasons for their exclusion were as follows: Fourteen studies did not focus on healthcare information [17–30]. One study did not focus on the age range of 14 to 49 years old [31]. Six studies presented evidence on general healthcare [7, 32–36]. Eight studies reported on sexual and reproductive healthcare services [37–44]. Fiteen studies reported on maternal healthcare services [45–55]. Three studies were literature reviews [3, 34, 56].

Following full article screening, there was an 80.77% agreement versus 68.64% expected by chance, which constitutes a good agreement between screeners (Kappa statistic = 0.39 and p-value <0.05). Besides, the McNemar's chi-square statistic suggests that there was not a statistically significant difference in the proportions of yes/no answers by reviewers with a p-value >0.05. Discrepancies between reviewers' responses following full article screening were resolved by involving a third reviewer.

**Characteristics of sources of evidence.** Table 2 depicts the characteristics of the included studies. All the eligible studies were published from the year 2004 to the present. The study setting for all the included studies was LMICs. All of the included primary studies showed evidence on access to healthcare information by women of reproductive age in LMICs. Two of the four included studies were conducted in rural settings [57, 58] and the other two were conducted in both rural and urban settings [59, 60]. The included studies were conducted in the following countries: One in Myanmar [59], one in South Africa [60], one in Eastern Uganda [58], and one in Nepal [57]. The total sample size from included studies was 11 134 participants and all were women. All studies focused on female participants [57–60]. Of the four included studies two were quantitative studies [58, 59], one was mixed methods [57] and one was a qualitative study [60]. One of the four included studies reported on universal health coverage [59], another reported on value for money of mobile maternal health information messages [60], two reported on increasing access to maternal healthcare services and exploring the role of telemedicine, respectively [57, 58]. Table 4 below indicates the results for individual sources of evidence.

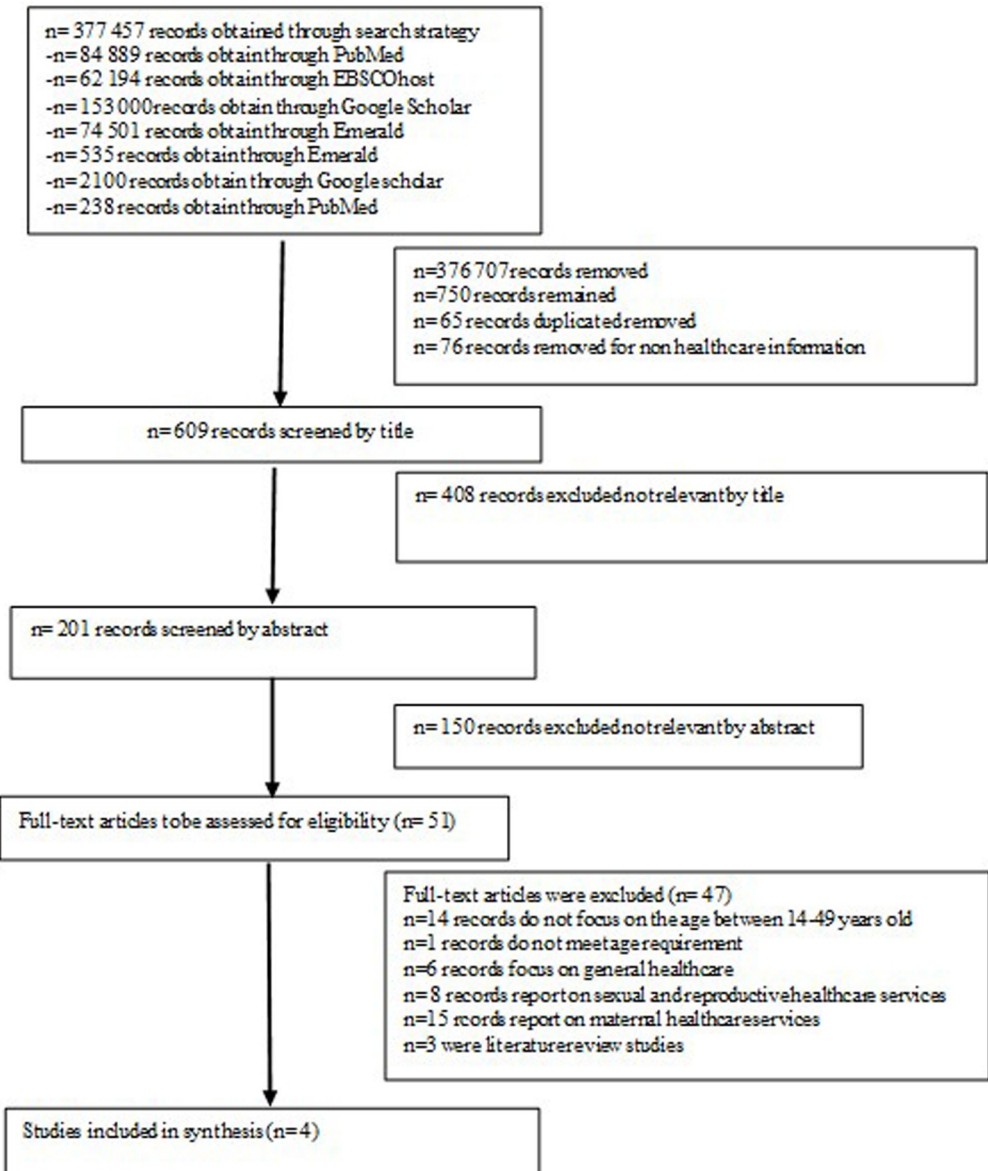

**Fig 1. Schematic diagram of the selection process for the studies used for the scoping review.**

## Critical appraisal within sources of evidence

All included studies underwent methodological quality assessment (additional file) using the MMAT version 2018 [16]. The overall percentage quality score was calculated for the included studies. The scores ranged from 71.4% to 100%. They were interpreted as follows: Below 51% low quality, 51–75% average quality, and 76–100% high quality.

## Synthesis of results

The themes that emerged from the four included studies were as follows: Accessibility, financial accessibility, connectivity, and challenges. All four studies showed evidence on access to healthcare information by women of reproductive age in LMICs.

**Table 4. Results for individual sources of evidence.**

| Author and date | Study title | Study design | Study setting (country) | Geographic setting (rural/urban) | Study population | Age | % of females |
|---|---|---|---|---|---|---|---|
| Han, 2018 | Progress towards universal health coverage in Myanmar | Quantitative stratified multistage design | Myanmar | Both | Demographic health survey (DHS) data and Integrated household living condition assessment. | Not indicated | Not indicated |
| LeFevre, 2018 | Forecasting the value for money of mobile maternal health | Qualitative-retrospective case control study | Gauteng, South Africa | Both | Pregnant women | 14–49 | 100 |
| Mayora, 2014 | Incremental cost of increasing access to maternal health | Quasi-experimental voucher study | Eastern Uganda | Rural | Two districts (three health sub-districts each) | 14–49 | 100 |
| Parajuli, 2017 | Exploring the role of telemedicine in improving access to healthcare services by women and girls in rural Nepal | Mixed method | Nepal | Rural | Girls and women | 17–37 | 100 |

**Accessibility.** Out of the four included studies, one study reported on the accessibility of maternal healthcare information to pregnant women. The study conducted in Myanmar reported on universal health coverage. Its main purpose was to determine the national and subnational health service coverage and financial risk protection [59]. Twenty-six health service indicators were examined by using nationally representative data from the Myanmar Demographic and Health Survey (2016) and the Integrated Household Living Condition Assessment (2010). The same study also assessed the incidence of catastrophic health payment and impoverishment caused by out-of-pocket payments [59]. The study findings showed that nationally, the coverage of health service indicators ranged from 18.4% to 96.2%. The findings further indicated that the coverage of most health services indicators did not reach the universal health coverage of 80% [59]. Also found is that increased levels of education (either the mothers or partners) have a positive influence on the access to perinatal care services [59]. The study showed that women with some higher education were likely to attend at least four ANC visits and to deliver in the health facilities compared to those with no education [59]. The study recommended that to achieve 80% coverage, efforts should focus on the expansion of services and increased coverage to reduce the gap that exists in healthcare information coverage [59]. Although the study mentioned the gap in maternal, neonatal, and health coverage, a research gap still exists to explore interventions that can be used to attract women of reproductive age to use healthcare information.

**Financial accessibility/affordability.** One study reported on the affordability of healthcare information to women of reproductive age. Mayora et al. (2014) conducted a study in eastern Uganda to assess the influence of demand- and supply-side programmes on increasing access to maternal health services [58]. This was a costing study that used vouchers. Costs were based on market prices as recorded in programme records [58]. Pregnant mothers were issued vouchers that they needed to present to the service provider, who, in return, would be reimbursed by the research team on presenting the voucher (Dec 2009 to March 2010 and June 2010 to June 2011) [58]. The outcome of this study revealed that transport vouchers scored the highest, followed by health system strengthening, while maternal services vouchers were the lowest [58]. The study further showed that the average cost of transport per women to and from the health facility was US$4.60 per woman. It also indicated that combined payment total incremental costs for delivery cost was the highest at US$ 317 157, followed by ANC at US$ 107 890, while postnatal care cost the least at US$ 7.60 [58]. These was a combined payment for both transport and service vouchers for all ANC sessions and delivery, health system strengthening, sensitization and mobilization and voucher administration. The findings also

revealed that when subsidizing maternal healthcare costs through demand and supply, side initiatives were a lesser cost and would require fewer resources than expected [58]. Although the study indicated that a voucher study can be used and may not require a significant amount of resources, an extensive research study still needs to be conducted on its sustainability and what would be a reasonable cost to the entire population.

**Connectivity.** Two studies reported evidence on the connectivity of women to healthcare information [57, 60]. A study conducted in Nepal reported on telemedicine for improving access to healthcare services by women and girls in rural Nepal [57]. The purpose of the study was to assess the influence of telemedicine in reducing gender-based challenges that women and girls faced to reach healthcare services in rural areas of Nepal [57]. The sample for this study were women and girls who used video conference-based telemedicine services before January 2015 and those who received mobile phone-based telemedicine in January of the same year [57]. The results of the study revealed that telemedicine positively influences travel restrictions, treatment expenses, and apprehension regarding sexual and reproductive health consultations. It was further shown that this intervention decreased travel time, because of timely assistance to access healthcare services for women and girls. At the same time, it created the convenience in improved time management so that the women would still be able to perform household chores and other activities [57]. This study revealed that telemedicine, especially mobile phone-based telemedicine, encouraged women and girls to have the confidence and freedom to ask about sexual health-related information from a doctor located at a far distance [57]. Besides, because of the confidentiality insurance of mobile phone-based telemedicine, fear, or timidity is reduced because the identity of the women and girls is not revealed [57]. A study conducted in Gauteng in South Africa that aimed at modeling the incremental cost-effectiveness of gradually scaling up text messaging services to pregnant women showed this as a cost-effective strategy for bolstering ANC and childhood immunizations [60]. The two studies indicated modalities of accessing services by women of reproductive age from a distance. However, there is still a research gap in how to access the population of women of reproductive age who are not within the diameter of network coverage and those who cannot afford to acquire technology devices such as cell phones.

**Challenges.** Two studies presented evidence on challenges or barriers to accessing maternal healthcare services by women of reproductive age [57, 59]. Study findings by Han et al. (2018) highlighted certain hindrances to effective implementation of maternal healthcare programmes, which included heavy workloads, geographical and transportation barriers, poor supervision and training, and insufficient replacement of auxiliary midwife kits [59]. The lack of accessible healthcare facilities, inadequate health workforce, and health budget allocation were the main causes of the regional inequity [59]. The main barriers in most Asia-Pacific countries, including Myanmar, are high use fees and cash payment for healthcare services, which are highly likely to hinder disadvantaged communities from accessing healthcare facilities [59]. Maternal, neonatal, and child health (MNCH) indicators such as postnatal care for neonates and institutional delivery recorded the lowest coverage [59]. Besides, financial constraints and lack of transportation are the prevalent hindrances to access delivery care facilities [59].

A study by Parajuli and Doney (2017) reported that even though telemedicine was inclined to reduce gender-based barriers for women and girls, we should take note that their capacity to benefit from telemedicine was limited, mainly in two ways. Firstly, women and girls who have no mobile phone found it difficult to call a remote doctor. Secondly, women with lower levels of education had to be assisted to utilize mobile phone-based telemedicine [57]. Although many challenges that may hinder women of reproductive age to access healthcare information have been outlined, a research gap exists on how these challenges can be alleviated to empower women of reproductive age to access maternal healthcare services.

### Risk of bias across studies

All studies scored between 71.4% and 100%. One of the included studies scored the highest quality score of 100% [59]. Two of the included studies scored 86% respectively [58, 60]. One study scored a quality score of 71.4% [57].

## Discussion

### Summary of evidence

This scoping review mapped the available literature on access to healthcare information by women of reproductive age in LMICs. It also provided a general overview of what factors contribute to women of reproductive age in LMICs not accessing healthcare information. The study revealed that that there was lack of literature in this area. Evidence provided information on the following themes: Accessibility, financial accessibility, connectivity, and challenges being faced by women of reproductive age to accessing healthcare services. It was also shown that women with high education have greater access to healthcare information than those with lower education. Furthermore, it was revealed that MNCH gaps exist. It also provided evidence on demand- and supply-side initiatives such as transport vouchers and maternal healthcare services vouchers. The findings of the study showed incremental cost-effectiveness of exposure to SMS text messages during the provision of maternal healthcare services, which may increase access to healthcare information. Evidence also showed that telemedicine reduces travel restrictions, treatment expenses, and apprehension regarding sexual and reproductive healthcare information.

The findings of this study showed that access to health services indicators was below the UHC, which, in turn, affects access to healthcare information. Similar to our findings, a study done in Nigeria by Aregbeshola et al. (2017) found that about 10–20% of the monthly household income was spent on healthcare by 46.8% of their respondents. It further found that a total of 97.9% of respondents had no health insurance coverage [33]. Another study conducted in Nigeria by Okoronkwo et al. (2015) found that the costs of medical treatment and not having insurance coverage was a major financial barrier to utilization and treatment services [61]. These two studies are in agreement with our findings in LMICs.

Similarly, a high-income country study conducted in New Jersey in the United States by Holstein et al. (2017) revealed that patient access to care under ten large insurance plans varied by the plan, but overall, access was difficult. Furthermore, maternal healthcare is said to be more accessible for women with high levels of education, and therefore have a greater chance of accessing healthcare information compared to those with low levels of education. This could be because educated women are generally more exposed to more information than non-educated women. A study by Vidler et al. (2016) found poor education to be one of the hindering factors in accessing maternal healthcare services [62]. Contrary to our findings, a study conducted in Pakistan in 2017 found that distance, transport, staff availability, income, service hours, and service organization are some of the barriers to maternal healthcare services where healthcare information can be accessed [63]. Furthermore, the existing MNCH gap that our study revealed could be due to the challenges or barriers that hinder access to maternal healthcare services in low research settings. Illustrating this, Asghari et al. (2018), in a study in urban slums of Lagos in Nigeria, found that a total of 80.3% of their respondents had an estimated travel distance ranging from 6 to 10 km to reach a healthcare facility [33]. Other barriers were revealed by a study conducted in Nepal by Paudel et al. (2018) where the low focus of primary healthcare on engagement and empowerment was responsible evident in two areas; firstly, in quality of care: poor acceptance, feeling unsafe and uncomfortable in health facilities; and

secondly, in health governance: failure in delivering healthcare services during pregnancy and delivery were some of the challenges identified [64].

Our study found that demand- and supply-side initiatives such as transport vouchers and maternal healthcare services vouchers were effective and may not require a significant amount of resources. However, contrary to our findings, a study conducted in Ghana on fee-free maternal healthcare services found that direct costs associated with ANC in the public healthcare facilities were still a significant barrier to pregnant women who wanted to utilize services from these facilities [46]. Contrary to our findings, a study conducted in India by Sahoo et al. (2017) found that service providers also experienced barriers that may hinder service provision, such as physical access to facilities [39]. Another study similar to ours conducted in Bangladesh by Wahed et al. (2017), on sex worker access to sexual and reproductive healthcare services, revealed that financial problems, shame about receiving care, the unwillingness of service providers to provide care, unfriendly behaviour of the providers and distance to care were some of the challenges that prevent them from receiving sexual and reproductive healthcare services [44]. Furthermore, the findings of these studies showed incremental cost-effectiveness of exposure to SMS text messaging during the provision of maternal healthcare services. This is in agreement with a 2016 systematic study finding that showed similar incremental cost-effectiveness of exposure to SMS text messaging during the provision of maternal healthcare services [6]. Our findings were also in agreement with a study conducted in Afghanistan by Yamin and Kaewkungwa (2018), showing that 81.7% of their participants were willing to receive health messages via a mobile phone. The same study also revealed that the automated voice call was the most preferred method for sending health messages. More than 90% of women are willing to receive reminders for their children's vaccination and ANC [65]. Evidence also showed that telemedicine reduced travel restrictions, treatment expenses, and apprehension regarding sexual and reproductive health. A high-income country study by Jandovitz et al. (2018) which was conducted with organ transplantation patients was in agreement with our findings that telemedicine has the potential to improve the healthcare delivery model by providing increased patient to healthcare team interactions and access, which optimizes engagement and outcomes [66].

The SDG target 3.7 focuses on the universal access to sexual and reproductive healthcare services, achievable by 2030 [67]. The WHO recommendations on health promotion interventions for maternal and newborn health stipulates twelve recommendations to strengthen maternal and newborn health. Two of the recommendations are interventions to promote awareness of human, sexual and reproductive rights and the right to access quality skilled care; and community mobilization through facilitated participatory learning and action cycles, highlighting women's groups to empower women with relevant information and knowledge to access healthcare services.

## Limitations

Our study findings show that there is limited published literature specific to strategies in place that would enable these women to access healthcare information in general in LMICs. The inclusion and exclusion criteria used in the study could be one of the limitation. Therefore, we hope that this study's results will prompt further studies to provide a contextual insight for these strategies to increase the use of maternal and reproductive healthcare services. Considering that only two studies mentioned interventions that can attract women of reproductive age to access maternal healthcare services in LMICs, it would have benefited by widening the inclusion criteria or narrowing the exclusion criteria. We would like to recommend future

pilot studies and randomized control trials to assess strategies aimed at enabling these women to access the healthcare information under discussion. We would like to further recommend that future intervention programmes should be developed and implemented to ensure quality and desirable maternal healthcare services outcomes. However, this study was limited in that we only included studies that were conducted between 2004 to the present, excluding those conducted before 2004.

## Strengths

This study encompassed examples of research undertaken in diverse settings such as rural, urban and semi-urban, which gives a clear view of the practical experiences and challenges that may be faced when accessing MCHI in other similar settings. Additionally, the full article screening tool was piloted, which resulted in improved reliability, as confirmed by the degree of agreement results; that there was 80.77% agreement versus 68.64% expected by chance constitutes a good agreement between screeners (Kappa statistic = 0.39 and p-value $<0.05$). Besides, the McNemar's chi-square figures indicate that there is no statistically substantial dissimilarity in the number of yes/no answers by the reviewer, with a p-value $>0.05$.

All primary studies incorporated underwent quality appraisal using an approved tool–the MMAT–to assess the methodological quality. The other important strength of this study is the fact that there was no limitation on language because it included studies written in other languages apart from English.

## Conclusion

This study demonstrated that some strategies might be useful to expose a large number of women from receiving MCHI without any challenges. It also indicated that there is a need for more research on evidence that would enable access to MCHI by women of reproductive age in LMICs.

## Supporting information

**S1 Table. PCC framework.**
(DOCX)

**S1 Fig. Evidence based-framework for access and utilization of maternal and child health information by adolescent girls during pregnancy.**
(DOCX)

**S2 Table. Results for individual sources of evidence.**
(DOCX)

**S3 Table. Database search.**
(DOCX)

## Acknowledgments

The authors would like to thank the supervisor Dr. Thompson-Mashamba for her inputs and technical support; the University of Kwazulu-Natal Postgraduate office which facilitated the protocol to BREC; the UKZN for the library facilities; and the UKZN Systematic Review Unit for training and technical support.

## Author Contributions

**Conceptualization:** Joyce Twahafifwa Shatilwe, Desmond Kuupiel, Tivani P. Mashamba-Thompson.

**Data curation:** Joyce Twahafifwa Shatilwe, Desmond Kuupiel, Tivani P. Mashamba-Thompson.

**Formal analysis:** Joyce Twahafifwa Shatilwe, Desmond Kuupiel, Tivani P. Mashamba-Thompson.

**Investigation:** Joyce Twahafifwa Shatilwe, Desmond Kuupiel, Tivani P. Mashamba-Thompson.

**Methodology:** Joyce Twahafifwa Shatilwe, Desmond Kuupiel, Tivani P. Mashamba-Thompson.

**Project administration:** Joyce Twahafifwa Shatilwe.

**Resources:** Joyce Twahafifwa Shatilwe.

**Software:** Joyce Twahafifwa Shatilwe.

**Supervision:** Tivani P. Mashamba-Thompson.

**Validation:** Desmond Kuupiel, Tivani P. Mashamba-Thompson.

**Visualization:** Desmond Kuupiel, Tivani P. Mashamba-Thompson.

**Writing – original draft:** Joyce Twahafifwa Shatilwe.

**Writing – review & editing:** Desmond Kuupiel, Tivani P. Mashamba-Thompson.

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
