## [Decision Letter · Decision Letter 0]

19 Oct 2020

PONE-D-20-25622

Evidence on access to health care information by women of reproductive age in Low-and-Middle-Income Countries: Scoping Review

PLOS ONE

Dear Dr. Shatilwe,

Thank you for submitting your manuscript to PLOS ONE. After careful consideration, we feel that it has merit but does not fully meet PLOS ONE’s publication criteria as it currently stands. Therefore, we invite you to submit a revised version of the manuscript that addresses the points raised during the review process.

The Reviewers have raised important concerns about the objectives and logical sequence of methods and results. Also, we note that the published protocol and this paper have a major overlap (similar sentences). Kindly rewrite those sections. Please review the manuscript for grammatical erros. 

We look forward to receiving your revised manuscript.

Kind regards,

Rohina Joshi

Academic Editor

PLOS ONE

Journal Requirements:

Reviewers' comments:

Reviewer's Responses to Questions

**Comments to the Author**

1. Is the manuscript technically sound, and do the data support the conclusions?

Reviewer #1: Yes

Reviewer #2: Partly

2. Has the statistical analysis been performed appropriately and rigorously? 

Reviewer #1: N/A

Reviewer #2: N/A

3. Have the authors made all data underlying the findings in their manuscript fully available?

Reviewer #1: No

Reviewer #2: No

4. Is the manuscript presented in an intelligible fashion and written in standard English?

Reviewer #1: Yes

Reviewer #2: No

5. Review Comments to the Author

Reviewer #1: This is a scoping review on access to health care information by women of reproductive age in LMICs. Four papers met inclusion criteria and the authors were able to discuss the following topics: accessibility, financial accessibility/affordability, connectivity and challenges. I suggest the authors review the entire manuscript for inconsistencies in grammar and typographical errors.

I have made comments below:

Major comments:

1. Introduction

a. The authors discuss healthcare service, maternal healthcare service and healthcare information in the first two pages of the manuscript. It is unclear to me what the focus of this scoping review is. I can see in the methods that healthcare information is one of the key words in the search strategy but access to information and access to service is drastically different. Also, all the results are maternal services. This manuscript would benefit if the authors streamline their terms.

b. The aim is stated on page 3 and objective on page 4--- please use only one of these.

c. This sentence on page 3 needs a reference: Women of reproductive age will utilize maternal healthcare information to their maximum if they have access to healthcare services.

2. Methods

a. Please reference your protocol rather than just linking the paper.

b. The long list that includes scoping review methodology (page 4, row 92-97) is unnecessary.

c. Add search strategy in appendix please.

d. Will need information on which authors performed the search, risk of bias, etc.

e. Add data availability statement

3. Results

a. Page 14, paragraph on financial accessibility- ‘The study further shows that the average cost of transport per women to and from the health facility was US$4.6. It further indicates that delivery cost was the highest with US$ 317,157 followed by antenatal care US$ 107 890 while post-natal care was the least with US$ 7.6’- are these costs per person? Please provide additional information.

b. This sentence on page 15 is too long. Consider revising. ‘The study further revealed that women and girls’ fear or timidity has been reduced because their identity is not being revealed because of the insurance of mobile phone-based telemedicine’ There are several sentences in the results and discussion section that are long and difficult to follow. Can the authors address this please?

c. Page 15 ‘The main barriers in most Asia-Pacific countries, including Myanmar, are high use fees and cash payment for health care services, which are highly likely to hinder disadvantaged communities from accessing healthcare facilities’. I am unsure if the authors can make this statement regarding most of Asia Pacific (bearing in mind that this is the results section and not the discussion) based on one study conducted in Myanmar. Perhaps better to move it to discussion if there is sufficient evidence.

4. Discussion

a. The second sentence under discussion has a question mark, is this meant to be a question?

b. While reading through this scoping review, one of the things that I thought was important to note was the lack of literature in this area. Consider mentioning this in the first paragraph of the discussion.

c. The first paragraph under limitations is not related to limitations.

Reviewer #2: Thank you for the opportunity to review this scoping review of evidence on access to health care information by women of reproductive age in low-and middle-income countries. While the concept is important, the research question and presentation of the methods, results and discussion are unclear and do not flow logically throughout the manuscript. For example, there are separate sections on objectives and main research questions and this is confusing, it would be valuable to clearly state what is meant by "evidence on access to health care information", and what type of health care information this scoping review was assessing access to. The key words searched are described in different sections of the methods section: i.e. the study population key words are listed under search strategy and additional key words around the types of studies are contained under the information sources heading, and again this is separate to the study inclusion and exclusion criteria. It would be helpful to better understand the specific inclusion/exclusion criteria the excluded articles did/ did not meet, particularly the reasons for exclusion of the final 47 articles. There are several references missing including to support statements made in the background "women of reproductive age will utilize maternal healthcare information to their maximum if they have access to healthcare services" and the risk of bias assessment tool used MMAT is also not referenced on first use and has been abbreviated (MMAT) each time used throughout the manuscript. The limitations section of the manuscript should be written to speak to the limitations of the study, for example the small number of studies that were included, would it have benefited by widening the inclusion criteria or narrowing the exclusion criteria. A large proportion of the manuscript has been previously published in the form of a scoping review protocol which may conflict with PLOS ONE publication requirements.

6. PLOS authors have the option to publish the peer review history of their article (what does this mean?). If published, this will include your full peer review and any attached files.

Reviewer #1: **Yes: **Dr Cheryl Carcel

Reviewer #2: No

---

## [Author Response · Author response to Decision Letter 0]

5 Nov 2020

Dear Editorial Manager 

Thank you for the opportunity granted to me to work on my manuscript. Attached kindly receive the revised manuscript and the manuscript with track changes. I inserted the in-text of table 2-4 as requested. Kindly see page 4, 5 and 12.

Below see my responses

Best regards

Joyce

Reviewers’ Responses

Introduction

a. The authors discuss healthcare service, maternal healthcare service and healthcare information in the first two pages of the manuscript. It is unclear to me what the focus of this scoping review is. I can see in the methods that healthcare information is one of the key words in the search strategy but access to information and access to service is drastically different. Also, all the results are maternal services. This manuscript would benefit if the authors streamline their terms.‬

Response: Terms such as healthcare service, maternal healthcare service and healthcare information, access to information and access to service has been streamlined. Please see line 21, 22, 34 and 48 as well as the background section. See page 2 and 3.

b. The aim is stated on page 3 and objective on page 4--- please use only one of these.

Response: Aim retained and section on objective removed. See page 3 and 4.

c. This sentence on page 3 needs a reference: Women of reproductive age will utilize maternal healthcare information to their maximum if they have access to healthcare services.

Response: Reference on the sentence inserted. See page 3.

2. Methods

a. Please reference your protocol rather than just linking the paper.

Response: Reference on the published scoping review protocol inserted. See page 4.

b. The long list that includes scoping review methodology (page 4, row 92-97) is unnecessary.

Response: Long list deleted. See page 4.

c. Add search strategy in appendix please.

Response: Search strategy added in appendix. See page 31.

d. Will need information on which authors performed the search, risk of bias, etc.

Response: Authors contribution inserted. See page 20.

e. Add data availability statement

Response: Data availability statement added. See page 21.

3. Results

a. Page 14, paragraph on financial accessibility- ‘The study further shows that the average cost of transport per women to and from the health facility was US$4.6. It further indicates that delivery cost was the highest with US$ 317,157 followed by antenatal care US$ 107 890 while post-natal care was the least with US$ 7.6’- are these costs per person? Please provide additional information.

Response: Additional information for cost financial accessibility added. See page 14

b. This sentence on page 15 is too long. Consider revising. ‘The study further revealed that women and girls’ fear or timidity has been reduced because their identity is not being revealed because of the insurance of mobile phone-based telemedicine’ There are several sentences in the results and discussion section that are long and difficult to follow. Can the authors address this please?

Response: Long sentences revised under on page. 15

c. Page 15 ‘The main barriers in most Asia-Pacific countries, including Myanmar, are high use fees and cash payment for health care services, which are highly likely to hinder disadvantaged communities from accessing healthcare facilities’. I am unsure if the authors can make this statement regarding most of Asia Pacific (bearing in mind that this is the results section and not the discussion) based on one study conducted in Myanmar. Perhaps better to move it to discussion if there is sufficient evidence.

Response: Paragraph deleted. See page 15.

4. Discussion

a. The second sentence under discussion has a question mark, is this meant to be a question?

Response: Question mark under discussion section removed, it was a typo error. See page 16.

b. While reading through this scoping review, one of the things that I thought was important to note was the lack of literature in this area. Consider mentioning this in the first paragraph of the discussion.

Response: A paragraph on ‘’Lack of literature lacking in this areas’’ has been inserted in the first paragraph. See page 16.

c. The first paragraph under limitations is not related to limitations.

Response: The first paragraph under limitations section which does not match with the section has been deleted. See page 18.

Responses for Reviewer #2

a) While the concept is important, the research question and presentation of the methods, results and discussion are unclear and do not flow logically throughout the manuscript. For example, there are separate sections on objectives and main research questions and 

Response: The method section has been re-arranged. The objective section and research questions has been deleted and aim of the study retained under background section as earlier mentioned under reviewer #1 responses.

b) This is confusing, it would be valuable to clearly state what is meant by "evidence on access to health care information".

Response: Insertion has been made with explanation as follows: This scoping review tried to search available interventions/strategies in place that enable women of reproductive age to access health care information. Please see page 3.

c) What type of health care information this scoping review was assessing access to.

Response: The scoping review was trying to search different interventions/strategies in place that enable women of reproductive age in low-and middle-income countries to access healthcare information. The interventions are such as healthcare promotion interventions programmes, health outreach programmes, facility based education initiative, health education initiatives (comprehensive sexuality education programmes), programmes to scale up healthcare information technology to promote technology (text messages, mobile health (M-health)), community based outreach programmes, school health programmes. Please see page 4 under materials and method section.

d) The key words searched are described in different sections of the methods section: i.e. the study population key words are listed under search strategy and additional key words around the types of studies are contained under the information sources heading, and again this is separate to the study inclusion and exclusion criteria. 

Response: Amendment made on page 5 under section search strategy and information sources. The key words were mainly guiding the database search process while also applying the inclusion and exclusion criteria. The inclusion and exclusion criteria are different from the key words although both are being used to guide the process of database search and the screening process. See page 5.

e) It would be helpful to better understand the specific inclusion/exclusion criteria the excluded articles did/ did not meet, particularly the reasons for exclusion of the final 47 articles. 

Response: Some of the reasons for exclusion were as follows: Fourteen records do not focus on the age between 14-49 years old, one records do not meet age requirement, six records focus on general healthcare, eight records report on sexual and reproductive healthcare service, 15 records report on maternal healthcare services and three were literature review studies. See page 9.

e) There are several references missing including to support statements made in the background "women of reproductive age will utilize maternal healthcare information to their maximum if they have access to healthcare services".

Response: References has been inserted under the phrase mentioned above, See page 3.

f) The risk of bias assessment tool used MMAT is also not referenced on first use and has been abbreviated (MMAT) each time used throughout the manuscript. 

Response: MMAT tool acronym has been spelt out in full and there after an acronym has been inserted. Please see page 7. 

g) The limitations section of the manuscript should be written to speak to the limitations of the study, for example the small number of studies that were included, would it have benefited by widening the inclusion criteria or narrowing the exclusion criteria. 

Response: Amendment has been made as follows: The first paragraph of under limitation section has been deleted and some insertions has been made. Please see page 18.

h) A large proportion of the manuscript has been previously published in the form of a scoping review protocol which may conflict with PLOS ONE publication requirements.

Response: Similarities in the two manuscripts (Scoping Review Protocol and Scoping Review Result Paper) has been addressed and revisions has been made accordingly. Amendments highlighted in purple colour.

In-text for the tables has been inserted on page 4, 5 and 12.

---

## [Decision Letter · Decision Letter 1]

16 Apr 2021

PONE-D-20-25622R1

Evidence on access to health care information by women of reproductive age in Low-and-Middle-Income Countries: Scoping Review

PLOS ONE

Dear Dr. Joyce Twahafifwa Shatilwe,

Thank you for submitting your manuscript to PLOS ONE. After careful consideration, we feel that it has merit but does not fully meet PLOS ONE’s publication criteria as it currently stands. Therefore, we invite you to submit a revised version of the manuscript that addresses the points raised during the review process.

Please read the attachment for recommendations to further improve the readability and flow of your manuscript for the benefit of readers.

Please submit your revised manuscript by April 30, 2021. If you would need more time than this to complete your revisions, please reply to this message or contact the journal office at plosone@plos.org. Please include the following items when submitting your revised manuscript:

We look forward to receiving your revised manuscript.

Kind regards,

Joyce Addo-Atuah, PhD

Academic Editor

PLOS ONE

Journal Requirements:

Reviewers' comments:

Reviewer's Responses to Questions

**Comments to the Author**

1. If the authors have adequately addressed your comments raised in a previous round of review and you feel that this manuscript is now acceptable for publication, you may indicate that here to bypass the “Comments to the Author” section, enter your conflict of interest statement in the “Confidential to Editor” section, and submit your "Accept" recommendation.

Reviewer #1: (No Response)

2. Is the manuscript technically sound, and do the data support the conclusions?

Reviewer #1: Yes

3. Has the statistical analysis been performed appropriately and rigorously? 

Reviewer #1: Yes

4. Have the authors made all data underlying the findings in their manuscript fully available?

Reviewer #1: Yes

5. Is the manuscript presented in an intelligible fashion and written in standard English?

Reviewer #1: Yes

6. Review Comments to the Author

Reviewer #1: Thank you for revising There are still abbreviations not properly spelled out such as MCHI in page 3.

7. PLOS authors have the option to publish the peer review history of their article (what does this mean?). If published, this will include your full peer review and any attached files.

Reviewer #1: **Yes: **Cheryl Carcel

---

## [Author Response · Author response to Decision Letter 1]

23 Apr 2021

Dear Editor 

We would like to tender our vote of appreciation to you and the reviewer team for attending to our manuscript despite this challenging time of COVID-19 pandemic.

Thank you very much for all the effort.

Joyce

---

## [Editor Report · Decision Letter 2]

27 Apr 2021

PONE-D-20-25622R2

Evidence on access to health care information by women of reproductive age in Low-and-Middle-Income Countries: Scoping Review

PLOS ONE

Dear Dr. Shatilwe,

Thank you for submitting your manuscript to PLOS ONE. After careful consideration, we feel that it has merit but does not fully meet PLOS ONE’s publication criteria as it currently stands. Therefore, we invite you to submit a revised version of the manuscript that addresses the minor points raised during the review process.

We look forward to receiving your revised manuscript.

Kind regards,

Joyce Addo-Atuah, PhD

Academic Editor

PLOS ONE

Journal Requirements:

Additional Editor Comments::

The resulting manuscript, which has taken into consideration all the reviewers' and editor's recommendations is a much improved version of the original one.

However, a few items need to be looked into again as follows:

1) Fig 1 is a Schematic diagram of the selection process for the studies used for the scoping review and should be titled as such because the current title of Fig 1 is not appropriate

2) Under the Discussion-Summary of Evidence---lines 6-8 The sentence is better stated as follows:

Women with high education have greater access to healthcare information than those with lower education.

3) Under limitations--line 9 on page 20

------randomized control trials to assess strategies aimed at enabling these women to access------ (note that the correct word to use before strategies is "assess" and not "access")

---

## [Author Response · Author response to Decision Letter 2]

27 Apr 2021

Dear Editor

Thank you very much for reviewing my manuscript. Your quick service is highly appreciated.

Kind regards

Dr. Shatilwe

---

## [Editor Report · Decision Letter 3]

30 Apr 2021

Evidence on access to health care information by women of reproductive age in Low-and-Middle-Income Countries: Scoping Review

PONE-D-20-25622R3

Dear Dr. Shatilwe,

We’re pleased to inform you that your manuscript has been judged scientifically suitable for publication and will be formally accepted for publication once it meets all outstanding technical requirements.

Kind regards,

Joyce Addo-Atuah, PhD

Guest Editor

PLOS ONE

Additional Editor Comments (optional):

The manuscript has been revised and updated taking into consideration all the recommendations including changing the title of Fig 1, however,Fig 1 has become too small in the latest manuscript
---

## [Editor Report · Acceptance letter]

14 May 2021

PONE-D-20-25622R3 

Evidence on access to healthcare information by women of reproductive age in low- and middle-income countries: scoping review 

Dear Dr. Shatilwe:

I'm pleased to inform you that your manuscript has been deemed suitable for publication in PLOS ONE. Congratulations! Your manuscript is now with our production department. 

Kind regards, 

on behalf of

Dr. Joyce Addo-Atuah 

Guest Editor

PLOS ONE